# K-Rod: An Innovative Method of Personalized Rib Splinting for Expeditious Management of Flail Chest in Acute Care Settings

**DOI:** 10.3390/medicina59010076

**Published:** 2022-12-29

**Authors:** Chia-Jung Lin, Yung-Sung Yeh, Yen-Ko Lin, Chao-Wen Chen

**Affiliations:** 1Division of Trauma and Surgical Critical Care, Department of Surgery, Kaohsiung Medical University Hospital, Kaohsiung 80756, Taiwan; 2Graduate Institute of Medicine, Kaohsiung Medical University, Kaohsiung 80708, Taiwan; 3Graduate Institute of Injury Prevention and Control, College of Public Health, Taipei Medical University, Taipei 11031, Taiwan; 4School of Post-Baccalaureate Medicine, College of Medicine, Kaohsiung Medical University, Kaohsiung 80708, Taiwan; 5Department of Emergency Medicine, Faculty of Medicine, College of Medicine, Kaohsiung 80708, Taiwan

**Keywords:** rib fractures, flail chest, rib splinting, K-Rod

## Abstract

Flail chest is a severe type of multiple rib fracture that can cause ventilation problems and respiratory complications. Historically, flail chest has been mainly managed through pain control and ventilatory support as needed. Operative fixation has recently become popular for the condition, and some studies have revealed its potentially positive effects on the outcomes of patients with flail chest. However, for those for whom surgery is unsuitable, few treatment options, other than simply providing analgesia, are available. Herein, we introduce our innovative method of applying personalized rib splinting for quick management of flail chest, which is easy, tailor-made, and has significant effects on pain reduction.

## 1. Introduction

For those with flail chest and stable hemodynamic status, pain is a major factor that exerts an adverse effect on respiratory function and quality of life [1]. Current treatment plans focus on surgical fixation and non-operative treatment. Surgical fixation tends to be more beneficial because it results in a shorter duration of mechanical ventilation use, shorter intensive care unit stay, and lower risks of acute respiratory distress syndrome and pneumonia [2]. Some studies reported surgery is associated with decreased hospital charges [3,4]. In Taiwan, the national health insurance (NHI) system covers most basic healthcare fees [5]. Surgical rib fixation is reimbursed but the plates and screws are out-of-pocket expense until mid-2021 [6]. For patients, higher healthcare expenses were incurred to the patients in the surgical fixation group [7]. For those who are not suitable or unwilling for surgery, nonoperative treatment might be considered as an alternative, which has a major disadvantage of prolonged pain. Although regional block such as epidural anesthesia, an intercostal nerve block, an intrathecal opioid, intrapleural analgesia, and a paravertebral block tends to be more effective than systemic pain control, it needs skin puncture [8,9,10].

A recent study has shown that rib splints have been designed to simultaneously reduce pain and increase lung capacity [11]. We aimed to find an easy, fast, and effective solution for the majority of patients with flail chest for whom surgery is unsuitable. The solution is named the “Kaohsiung Medical University Rib Orthotic Device” (K-Rod), which is easily constructed using common materials. It incorporates a three-dimensional (3D) computed tomography (CT)-based assessment, a tailored design, and a portable application. The K-Rod is described in detail in the following.

## 2. Technique

A 42-year-old man presented with severe chest pain after being involved in a fall. On arrival, he had clear consciousness, stable blood pressure, and slight tachypnea. Chest CT scan revealed overt left second to eighth rib fractures, and the fourth to eighth ribs were broken in two places, as flail chest is defined. Because of severe pain (visual analog scale (VAS) score: 9 points), he complained of shortness of breath, and supplemental oxygen was provided with a measured SpO2 of 90%. Oral and intravenous analgesics were applied, but his chest pain was intractably controlled during the first 24 h. Operation of rib fixation was proposed after evaluation by a thoracic surgeon, but he turned down the proposal because the out-of-pocket expense of surgery was unaffordable. We decided to apply the K-Rod. A reformatted 3D scan of the chest CT scan was acquired to evaluate fracture sites in all involved ribs (Figure 1).

As the locations of fracture sites were crucial to the effectiveness of rib splinting, we inspected and palpated his torso in detail and then marked tender points on the basis of 3D images of his chest. Subsequently, a handmade K-Rod was placed that covered all fracture sites and adjacent stable ribs. The K-Rod comprised four layers (Figure 2).

The first (innermost) layer was adhesive hydrocolloid dressing, which protected the skin and acted as an interface, providing stability between the splint and skin. The second layer was a double-sided foam tape applied to bridge the hydrocolloid dressing and hard part of the splint. The third layer, made of six-layered fiberglass casting tape (3M Scotchcast, 4 inches wide), was the only hard part of the K-Rod. The casting tape was trimmed and tailored for the complete coverage of injury sites. It was immersed in room-temperature water and then immediately adhered to the chest wall tightly before it hardened. Finally, the fiberglass layer of the splint was securely fixed over injury sites by using elastic adhesive tape, which formed the fourth (outermost) layer (Figure 3).

Using this K-Rod for external fixation of fractured ribs, an expeditious and excellent result was achieved. The patient’s pain–inspiration–cough (PIC) score before and after splinting was 3 and 7. His pain was greatly alleviated, as indicated by a postprocedural VAS of 4 points, and he could ambulate on his own immediately. The next day, he stopped using supplemental oxygen and had an improved pain score (VAS: 1 point). We provided only oral acetaminophen three times daily as supplemental pain control. His PIC score improved to 9 points in the following 48 h, and he was uneventfully discharged after a 6-day stay.

From March to December 2020, we applied K-rod to 43 patients, pain score (VAS) significantly decreased after the application of the K-Rod from a median of 8/10 to 2/10 (Table 1). Incentive spirometry was also used for these patients as a tool for lung function evaluations and risk of pulmonary complication prediction for those with fractured ribs [12]. In our study, patients achieved a higher volume of incentive spirometry immediately after the K-Rod was applied (from 510 mL to 750 mL).

## 3. Discussion

In recent years, operative management of flail chest has gained popularity and proved with benefits [2,7,13,14,15,16]. Surgical rib fixation provides significant pain reduction, but it is actually performed for only a limited proportion of patients. For the majority of patients treated nonoperatively, optimal pain control is paramount [17]. Our innovative method can be easily applied in nonoperative patients with flail chest and offers excellent pain control.

Rib splints have been proposed previously for relieving pain and increasing vital capacity, particularly in prehospital settings and when analgesia is inadequate. However, splinting ribs can be challenging because of their diverse location and complex severity. Furthermore, a circumferential bandage exerting compressive force over the chest restricts thoracic movement and prevents adequate ventilation. A previous study has shown the effectiveness of fixation through chest orthosis or rib splints [11]. Uncustomizability, in addition to cost and accessibility, is the major weakness of this approach.

The K-Rod demonstrated several strengths. First, postprocedural pain relief is promptly achieved. Second, our rib splint is personalizable and scalable. By evaluating 3D CT scans, the fracture location and pattern were clearly recognized. Furthermore, by localizing tender points through palpation, we were confident regarding the size and shape of the rib splint required prior to splinting. Different from the previous use of circumferential rib belts, our tailored splint was employed to merely fix the injury site, having limited effect on lung expansion. Furthermore, because of the superior structural stability of the flail site compared with that achieved in other approaches, our patient had higher incentive spirometry performance after rib splinting. Lastly, our method has several other advantages including portable intervention, quick application, and lower cost compared with surgical fixation. The only challenge we had encountered before was inadequate fixation because fractured ribs were just under a woman’s large breast or scapula.

## 4. Conclusions

In summary, the K-Rod can be applied in an acute care setting and has numerous clinical and economic merits. Our method would have a vital positive effect on the current practice of flail chest management under limited resources.

## Figures and Tables

**Figure 1 medicina-59-00076-f001:**
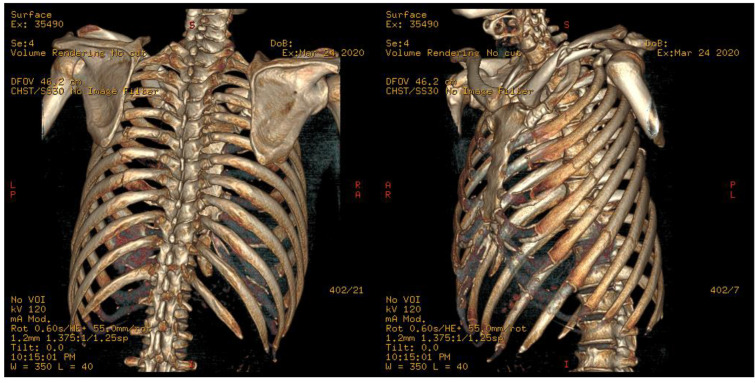
3D Chest CT scan depicting multiple fractures involving left middle clavicle and left 2nd–8th ribs. Flail fragments were noted over the 4th–8th ribs.

**Figure 2 medicina-59-00076-f002:**
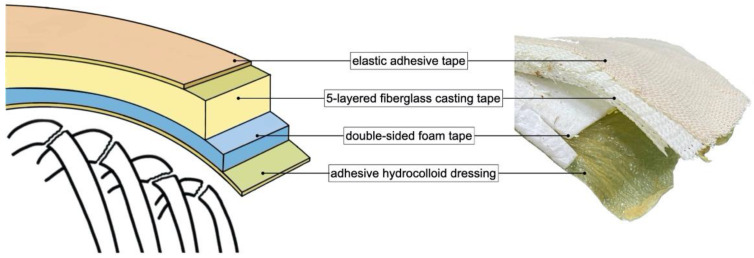
Cross-sectional view of a K-Rod. The four layers of the K-Rod are as follows: an innermost layer of adhesive hydrocolloid dressing, a layer of double-sided foam tape, a piece of five-layered fiberglass casting tape, and an outmost layer of elastic adhesive tape.

**Figure 3 medicina-59-00076-f003:**
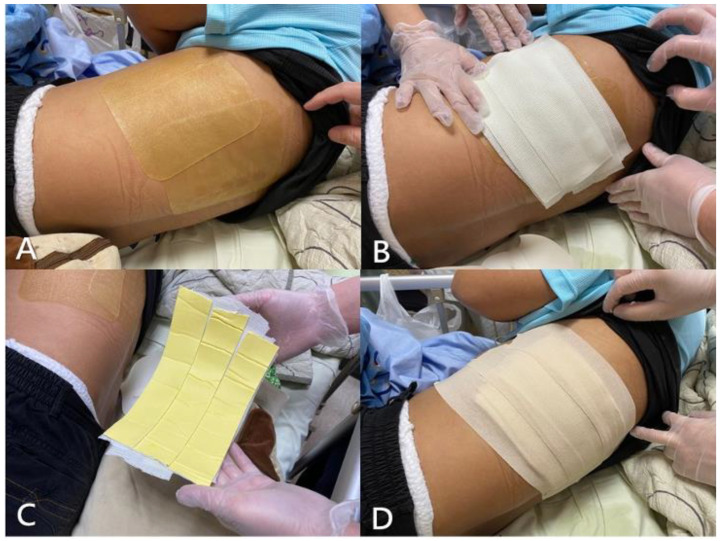
Steps of applying the K-Rod: (**A**) The innermost layer of adhesive hydrocolloid dressing was attached first to protect the skin and with complete coverage of all recognized tender points. (**B**) A six-layered hard splint made of fiberglass casting tape was trimmed and then immersed in water. It was then firmly placed over the injury area for shaping and hardening. (**C**) Once the fiberglass casting tape had hardened, it was detached from the chest wall, and one layer of double-sided foam tape was applied to its inner side. (**D**) The fiberglass casting tape was adhered to the innermost layer by using double-sided foam tape, and all layers were securely fixed using an outermost layer of elastic adhesive tape.

**Table 1 medicina-59-00076-t001:** VAS score measured immediately before and after K-Rod application.

VAS Score	Mean (SD)	Median (IQR)	*p*-Value *
Before	7.38 (2.11)	8 (7–9)	<0.001
After	2.05 (1.92)	2 (0–3)	

* Wilcoxon test.

## Data Availability

The data presented in this study are available on request from the corresponding author.

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
