# Peer review of "K-Rod: An Innovative Method of Personalized Rib Splinting for Expeditious Management of Flail Chest in Acute Care Settings"

_medicina, 2022, doi:10.3390/medicina59010076_

Round 1
Reviewer 1 Report
Introduction:
- I would check you statement for accuracy "higher health care expenses were incurred in the surgical fixation group." Yes, the surgery does cost money, but the fact these patients get off the vent sooner, out of the ICU sooner and therefore out of the hospital sooner results in decreased costs. There are several papers showing this so I would update this sentence (Akash Bhatnagar, MD, John Mayberry, MD, FACS, Ram Nirula, MD, MPH, FACS. JACS. 2012; Julia R. Coleman, MD, MPH, Kiara Leasia, MD, Ivor S. Douglas, MD, Patrick Hosokawa, MS, Ryan A. Lawless, MD, Ernest E. Moore, MD, andFredric Pieracci, MD, MPH JTACS. 2020)
Material and Methods:
- The word "evaluated" in line 40 should be changed to "evaluation"
- Elaborate more on why the surgery was "unaffordable." I am assuming things are different in different countries so please clarify why the surgery was unaffordable so all readers can relate.
Discussion:
- The first sentence in your discussion needs to be corrected. There have been four prospective randomized trials clearly showing superiority of rib fixation over non-operative management for flail chest with great benefits. To say there is only limited evidence available is quite inaccurate. Please correct (1. Tanaka H, Yukioka T, Yamaguti Y, et al. Surgical stabilization of internal pneumatic stabilization? A prospective randomized study of management of severe flail chest patients. J Trauma. 2002;52(4):727-32; discussion 732. 2. Marasco SF, Davies AR, Cooper J, et al. Prospective randomized controlled trial of operative rib fixation in traumatic flail chest. J Am Coll Surg. 2013;216(5):924-32. 3. Granetzy A et al. Surgical versus conservative treatment of flail chest. Evaluation of the pulmonary status. Ineractive Cardiovascular and thoracic surgery. 2005. 4. Pieracci, F, Yihan L, Rodil M, et al. A prospective, controlled clinical evaluation of surgical stabilization of severe rib fractures. J Trauma. 2016;80 (2) :187-194. )
- Can you elaborate more on the creation of splint and the difficulties you encountered or areas you could see as potential difficulties? You touched briefly on it at the end but a couple of questions. What if the fractures are really anterior or posterior? Is this just for lateral fractures? What if the patient has broken 10 or 12 ribs on that side? How many fractures do you have to cover with the splint to be effective? Do you have to cover both fractures of a rib with the splint in flail chest patients? What about rib healing? Do the fractures heal or do these patients end up with non-unions? Does their chest walls expand well or this there chest wall caving in from the flail pattern and therefore long term issues with SOB? Have you done any long term follow-up with this yet? If so please include it?
- How do the authors think the splint truly works to provide these results? Even if it is theoretical, I think it would add value to the manuscript.
Author Response
Please see the attachment, thank you.

Reviewer 2 Report
In abstract Author must be descript about the novelty of concepts involved, the validity of the technique and its potential for clinical applications
Author can be modified the manuscript with rules write Technical notes should consist of the following headings: short abstract (structured or unstructured), brief introduction, methods, results and discussion. The methods and results sections may be combined under the heading of “technique”. A technical note is a type of article that describes a specific technique or procedure, a modification of an existing technique, or a new equipment applicable to medicine. This type of article also covers technical innovations and developments. The technique should be well-tested and should preferably have some practical value in the clinical setting. Most importantly, a technical note should be written concisely and clearly.
Manuscript must be concist advancement on techniques and devices already available
Author Response
Please see the attachment, thank you
